# Prediction of urban heat island intensity based on multiple linear regression and deep learning

**Xinran Liu, Xia Zhu** 📵 *, **Yuanping Liu, Cui Jia, Jie Cao, Yu Zhong, Yu Zhang, Yunjie Zhang**

Department of Remote Sensing Information Engineering, North China Institute of Aerospace Engineering, Lang fan, He Bei, China

* zhuxia1201@nciae.edu.cn

## Abstract

The rapid urbanization process has led to many prominent environmental issues in urban areas, resulting from a drastic change in land use. The Urban Heat Island (UHI) effect is of particular concern because it has a significant impact on the livability of cities. Therefore, exploring and studying the intensity of UHI and its future distribution have significant practical implications. In this study, the CNN-LSTM-Attention model was constructed to predict four remote sensing spectral indices, and combined with the CA-Markov model to predict land use change. The relationship between four different spectral indices and the intensity of UHI was analyzed by a multiple linear regression model ($R^2 = 0.7468$, RMSE $= 0.0546$), and the UHI intensity and distribution in 2025 were predicted and analyzed. The results show that by 2025, the proportion of built-up area will continue to increase by 1.37 percentage points, which will lead to a more intense and concentrated UHI effect, and the proportion of heat island area will increase by 2.97 percentage points. The study shows that increasing vegetation area and water area can effectively alleviate the impact of UHI. Local government departments formulate reasonable policies based on survey results, reduce over-radiation, improve urban livability, and promote sustainable development.

## Introduction

Addressing the UHI effect is crucial for enhancing urban livability and promoting sustainable development. Various planning strategies have been proposed to mitigate this phenomenon. One such approach is "Compact Cityism," which aims to create high-density, mixed-use environments that reduce urban sprawl and associated heat-trapping infrastructure. By concentrating development in specific areas, cities can minimize the expansion of impervious surfaces and optimize public transportation systems, thereby decreasing energy consumption and heat generation.

Another strategy is "Green Wedge Urbanism," which involves integrating green spaces such as parks, forests, and wetlands into urban layouts in a wedge-shaped

**Data availability statement:** All relevant data are within the paper and its Supporting Information files.

**Funding:** The author(s) received no specific funding for this work.

**Competing interests:** The authors have declared that no competing interests exist.

pattern. These green wedges extend from the urban fringe to the city center, providing pathways for cool air to circulate and reducing the heat island effect. They also enhance biodiversity, improve air quality, and offer recreational spaces for residents.

Additionally, "Low-Carbon Urbanism" focuses on reducing greenhouse gas emissions through sustainable energy use, energy-efficient building designs, and the promotion of non-motorized transportation. By decreasing the overall carbon footprint of urban areas, this approach contributes to lower urban temperatures and a more sustainable urban environment.

These planning strategies offer valuable insights for urban planners and policy-makers in their efforts to combat the UHI effect and improve urban living conditions.

Global urbanization is occurring at a rate never seen before, with 6.3 billion people predicted to live in cities by 2050. Urbanization, the most significant human activity on Earth's surface, is responsible for the transformation of natural surfaces, particularly vegetation and permeable areas, into artificial surfaces. This shift results in significant changes to land cover and use, which applies to both urban and rural areas. The cyclical feedback process between urbanization and the climate environment involves the expansion of cities, which can have a significant impact on meteorological parameters in the region. Climate and environmental issues, represented by the UHI effect, have gradually evolved into major environmental problems in urban areas.

The "UHI effect" refers to the phenomenon of higher temperatures in urban areas than in the surrounding suburbs [1,2]. The impact of human activities, particularly urbanization, on the local climate is significant [3]. Understanding the dynamic changes in land use is essential to comprehending the effects of human activity on the environment [4,5]. Severe UHI effects can lead to increased environmental pollution [6]and a sharp rise in resource consumption [7], and serious threats to human health [8]. With the deepening of global changes, the analysis of land use changes has progressively evolved as a central theme in studies on global environmental change. Following the analysis of satellite data on land dynamics worldwide between 1982 and 2016, Song discovered that changes in land usage had regional features; For example, tree cover has increased in mountainous locations, but vegetation cover has decreased in many arid and semi-arid ecosystems (such as those in Australia, China, and the United States). Research by Mohammad Harmay and colleagues showed that from 2001 to 2014, the urban heat island intensity(UHII) increased uniformly, with an increase of 1.20±0.20°C, and the expansion of built-up areas reached 14.93% [9]. Kalnay's research projected an average surface temperature rise of 0.27°C per century [10]. Tian's study on the connection between UHI and urban livability [11]. Guo's studies on the connection between UHI and sustainable development have both produced insightful findings [12]. The ongoing escalation of the UHI phenomenon has resulted in notable adverse effects on the sustainability and livability of metropolitan areas, drawing widespread attention. Many different mixed land use factors influence the UHI effect [13], the built environment [14], anthropogenic heat from human activities [15,16], and other urban factors [17]. These correlations have been quantitatively studied using a variety of statistical techniques. LST is among the most significant indicators of the UHI effect [18]. A number of

studies have investigated the relationship between LST and land use using historical remote sensing data [19,20]. They have reached similar conclusions that the LST is higher in urban areas [21], while it is lower near water bodies and forests [22,23].

The prediction of land surface temperature today uses a wide range of spatial prediction models and techniques, including CA-Markov and artificial neural networks. However, these studies only rely on historical LST data and land use classification datasets to predict LST, without thinking about the future changes in land use. Such predictions may lead to reduced accuracy. In order to overcome this limitation, Nadizadeh Shorabeh used a combined model of CA-Markov and artificial neural networks to predict the UHI effect's strength in 2033 based on changes in land use [24]. Some researchers have also incorporated spectral indices into prediction models, such as Normalized Difference Vegetation Index (NDVI), Urban Index (UI), and Normalized Difference Built-up Index (NDBI) [25–27]. Wang used the link between land use, NDVI, LST to estimate the LST in Nanjing in 2030 and 2050 [28]. UI was selected as an LST predictor in a Mushore predictive research carried out in Harare. They projected that between 2015 and 2040, the portion of the city with LST between 18 and 28°C would shrink, while the portion with LST between 36 and 45°C would rise from 42.5% to 58% [29]. However, this research process did not fully consider the influence of land use alterations on urban LST. These studies also cannot directly serve as guidance for local governments to formulate urban planning policies to weaken the UHI. In this study, combined with the spectral indices of the research area, a multiple linear regression(MLR) model is constructed based on fully considering changes in land use, extracting the representative spectral indices of the research area according to land use classification(LUC) levels, to predict the UHI effect in the research area.

Beijing, the People's Republic of China's capital, has experienced a sharp increase in urbanization in the past few years. The UHI has been exacerbated in Beijing due to its massive expansion, which has led to extreme high temperatures multiple times during summers. The livability of Beijing is facing unprecedented challenges. Currently, China is continuously promoting ecological civilization construction, and a large number of ecological and environmental issues represented by the UHI effect will receive greatly increase attention from the public. Research on Beijing, China's capital, will be a valuable resource for other rapidly developing cities.

Based on the above comprehensive judgment, this paper plans to forecast the distribution and intensity of the urban UHI in Beijing in 2025 based on land use classification and spectral indices. The main objectives include: (1) to examine how land use changes and LST patterns have changed over time in Beijing between 2013 and 2022; (2) to construct models for predicting spectral indices and MLR models; (3) to simulate and predict the UHII and distribution of the UHI in Beijing in 2025.

## Materials and methods

### Study area

Beijing is bordered to the east by Tianjin Municipality and the west by Hebei Province. It serves as China's political, cultural, economic, and technological center. Since the initiation of reforms and opening up in 1978, Beijing has undergone rapid urbanization. According to statistics, the urbanization rate of Beijing, i.e., from 77.5% in 2000 to 87.5% in 2020, the percentage of people living in cities grew (Beijing Municipal Government, 2021). By 2023, the GDP of Beijing reached 4376.07 billion yuan, a 5.2% rise from the prior year. Calculated according to the resident population, the per capita GDP of the entire city was 200,000 yuan. Beijing successfully hosted the Summer Olympics and the Winter Olympics, making it the first "Dual Olympic City" globally. In addition, it is a prominent ancient capital, one of the birthplaces of the Chinese people, and a contemporary global city. Beijing has a clear four-season climate, semi-humid and semi-arid, with relatively short spring and autumn. This study selected six districts in the center of Beijing and some areas around them as the research area.As shown in Fig 1.

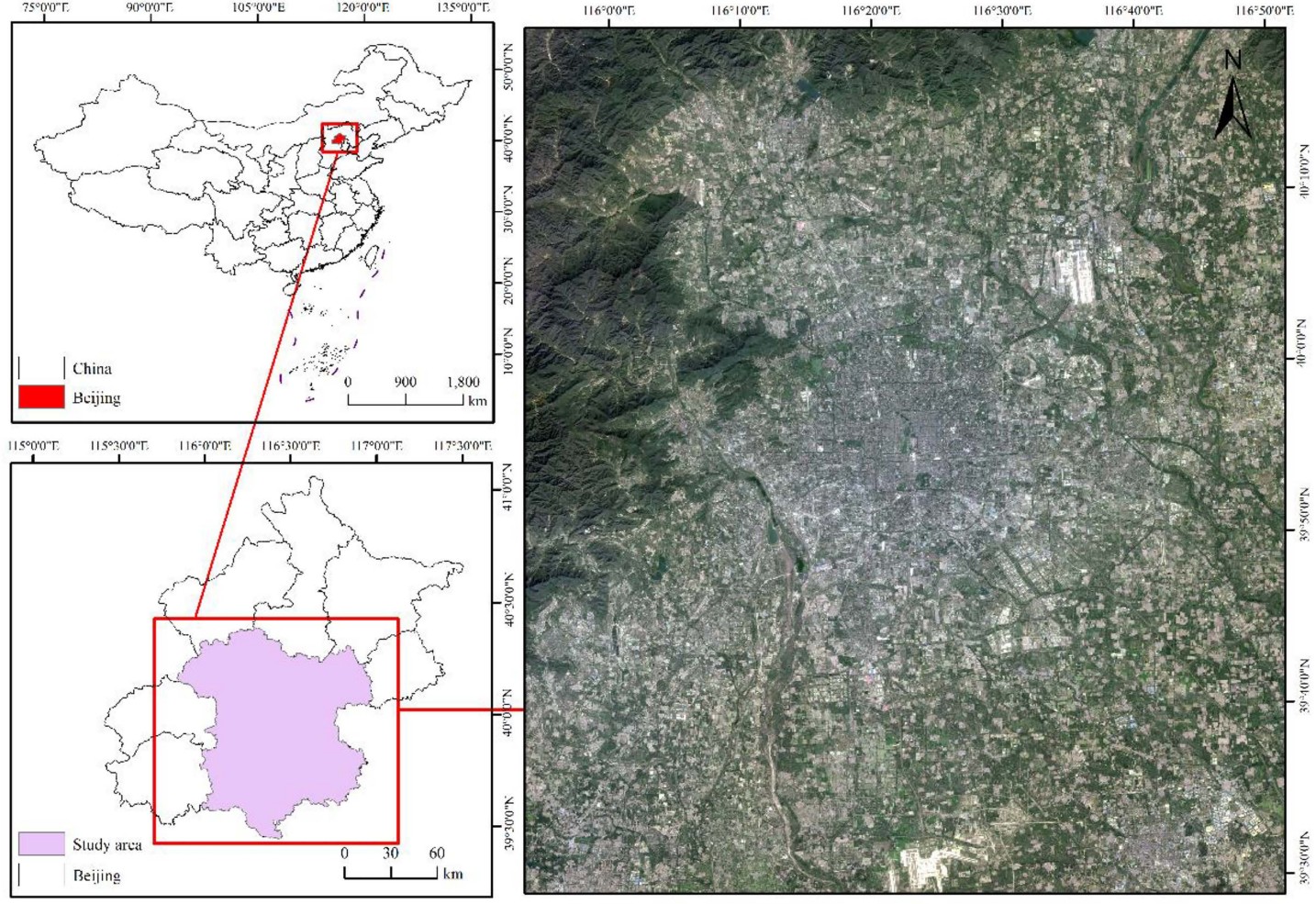

**Fig 1. The location of the study area. These include a map of China, a map of Beijing city, and a Landsat 8 true-color composite image of the study area.**

## Data collection and preprocessing

**Calculating landsat images for UHI.** Acquiring Landsat C2L2 dataset with cloud cover less than 10% from the website https://earthexplorer.usgs.gov. Since the data stored on the website are in integer format, there may be certain scaling factors and offsets in the actual results. Referring to the data user manual provided by the US Geological Survey website at https://www.usgs.gov/media/files/landsat-8-collection-2-level-science-product-guide, the Band Math function in ENVI software can be used to calculate true surface temperature by applying the formula B1*0.00341802 + 149. This will yield surface temperature in Kelvin units. To convert it to Celsius, subtract 273.15.

**Landsat images for land use classification.** This study examines the alterations in land cover area within the designated region from 2013 to 2022, utilizing four Landsat images sourced from the USGS website, and the cloud cover in each image area is less than 10%. The selected images were acquired between June and September to effectively mitigate the effects of seasonal variations. To enhance image quality and minimize atmospheric and lighting influences on surface reflectance, radiometric and atmospheric corrections were performed using ENVI 5.3.

## Technical workflow diagram

The goal of this study is to predict the intensity and distribution of the UHI effect. In order to fulfill this goal, the following workflow (Fig 2) is designed, consisting of 6 main steps: (1) Calculate the intensity of the UHII and LUC based on Landsat images. (2) Compute four remote sensing indices. (3) Establishing a Long Short-Term Memory Neural Network Model Based on the Attention Mechanism and Combined with Convolutional Neural Networks (CNN-LSTM-Attention) and a MLR model. (4) Predict LUC using the CA-Markov model. (5) Forecast temporal remote sensing indices using CNN-LSTM-Attention. (6) Predict future UHII and its distribution range based on the MLR model.

## Calculate urban heat island intensity

UHII is a scalar quantity used to characterize the strength of the UHI effect. It can be represented by the surface temperature difference between urban and suburban areas retrieved from satellite remote sensing. According to the local standard "UHII Classification" issued by the Beijing Municipal Administration for Market Regulation on March 30, 2023, the calculation method for UHII is as follows:

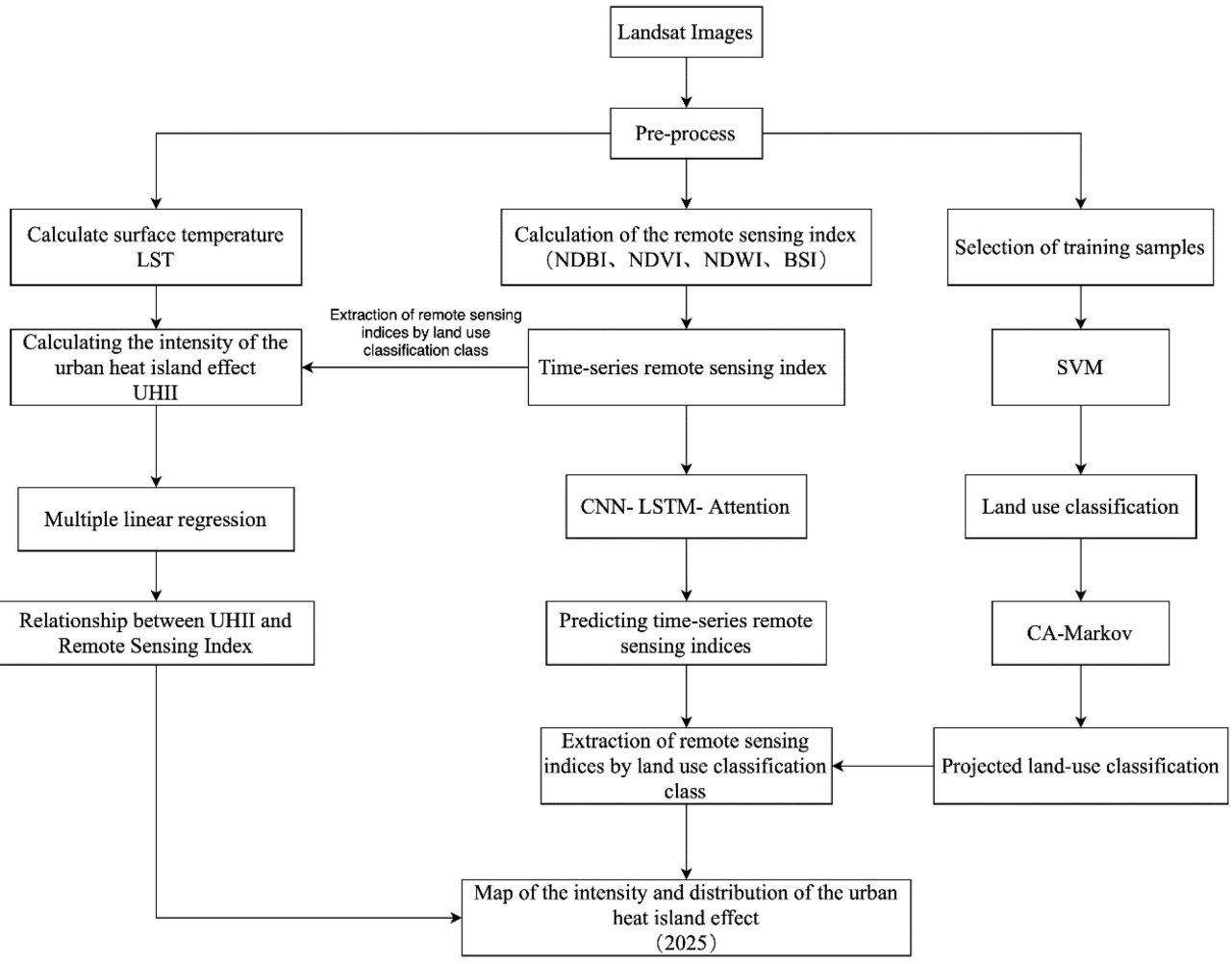

**Fig 2. Technical workflow diagram.**

$$UHII = T_u - T_r \tag{1}$$

where UHII represents the intensity of UHI; $T_u$ represents urban air temperature (°C) or surface temperature (K); $T_r$ represents suburban air temperature (°C) or surface temperature (K). In this article, based on the LUC map, suburban areas are defined as all places other than built-up regions and water bodies.

## Land use classification

The article compares three algorithms: Support Vector Machine, Minimum Distance, and Maximum Likelihood Classifier, and selects Support Vector Machine with the highest accuracy as the image classification algorithm. Compared with other algorithms, Support Vector Machine has lower requirements for training data. The Kernel Function Chosen is the Radial Basis Function; the value of Gamma is 2; the value of the penalty parameter is 200. In this article, four main land use types in Landsat images are selected: built-up areas, vegetation areas, water areas, and bare soil areas. By randomly selecting 1000 points from every categorised map and contrasting them with Google Earth historical images, accuracy verification is completed. The classification accuracy for 2013, 2016, 2019, and 2022 is 86.22%, 84.61%, 85.12%, and 82.37%, respectively.

## Compute remote sensing spectral indices

After reviewing relevant literature, several kinds of metrics of land cover with proven potential for predicting LST have been identified [30–34]. From these, the most representative remote sensing indices for the four land use categories were selected as the land cover indices for this study. By integrating spectral bands and constants, four land cover index indicators were computed for the years 2007, 2010, 2013, 2016, 2019, and 2022. These are the NDBI, NDVI, Normalized Difference Water Index (NDWI), and Bare Soil Index (BSI).

**Calculate NDBI.** NDBI is used in remote sensing to pinpoint built-up areas. It achieves rapid and accurate extraction of urban building information by comparing the reflectance differences between built-up regions and other types of land cover.

$$NDBI = \frac{SWIR-NIR}{SWIR+NIR} \tag{2}$$

where SWIR represents the reflectance in the shortwave infrared band; NIR represents the reflectance of the near-infrared band.
The time-series NDBI indices of the study area over the past decade have been calculated through the Google Earth Engine (GEE) platform, as shown in Fig 3 Overall, the NDBI shows a trend of periodic variation.

**Calculate NDVI.** Plants strongly reflect green and infrared light. Higher levels of vegetation cover result in lower red reflections and larger near-infrared reflections.. Near-infrared reflection rises with vegetation, but red light easily reaches saturation absorption. Therefore, by utilizing the difference between near-infrared and infrared, the contrast between infrared and near-infrared is enhanced, serving as an indicator reflecting vegetation. The expression for NDVI is as follows:

$$NDVI = \frac{NIR-Red}{NIR+Red} \tag{3}$$

where Red corresponds to band 4 in Landsat 8, representing the reflectance of the red band.
The time-series NDVI indices of the study area over the past decade have been calculated through the GEE platform, as shown in Fig 4. Overall, the NDVI shows a trend of periodic variation.

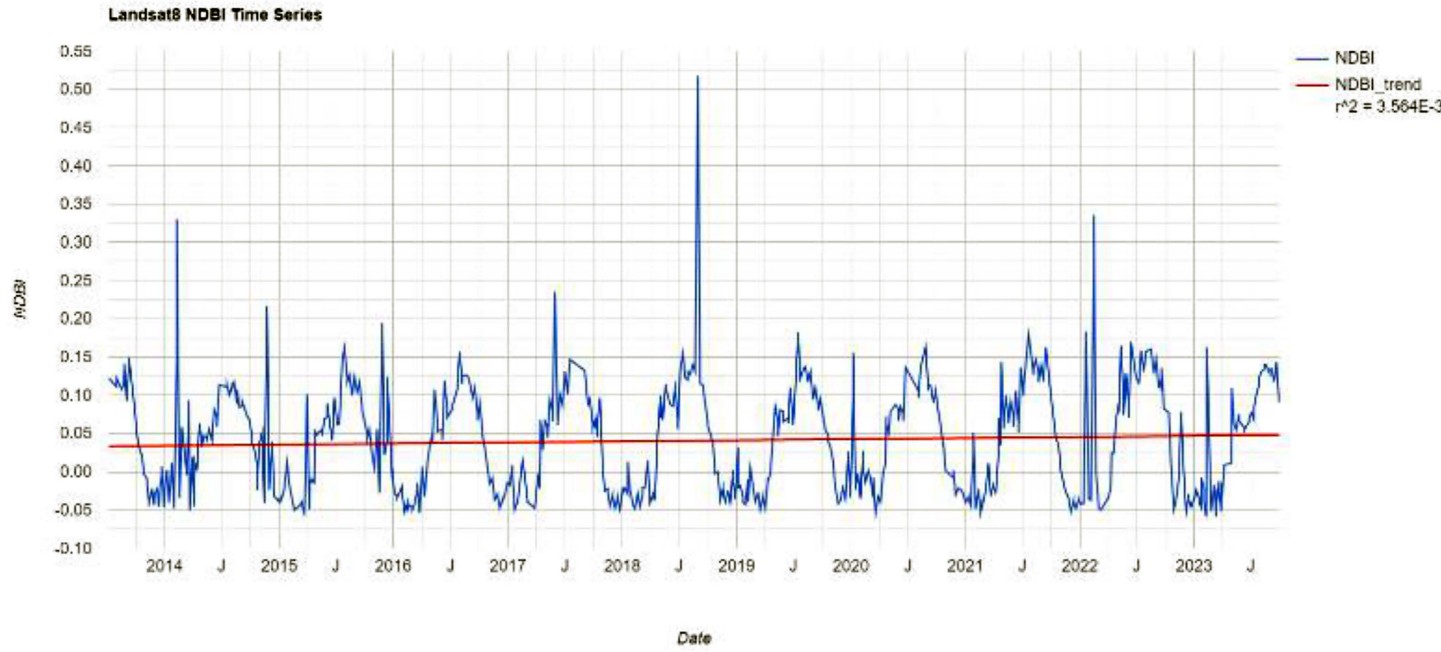

**Fig 3. The time series information of the NDBI.**

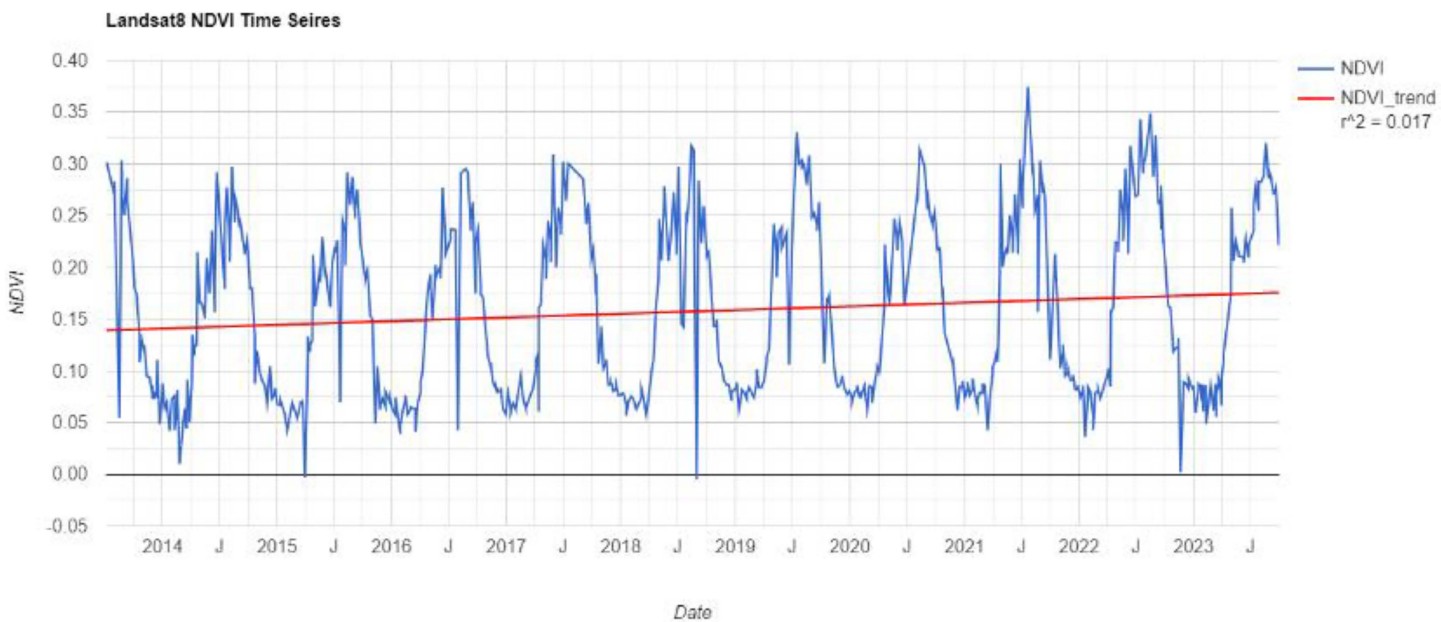

**Fig 4. The time series information of the NDVI.**

**Calculate NDWI.** NDWI is the most commonly used index for water extraction. Due to the decreasing reflectance of water with increasing wavelength, NDWI utilizes the Green and NIR bands for normalization and can effectively extract water information. The expression for NDWI is as follows:

$$NDWI = \frac{Green - NIR}{Green + NIR}$$

(4)

where Green corresponds to band 3 in Landsat 8, representing the reflectance of the green band.

The time-series NDWI indices of the study area over the past decade have been calculated through the GEE platform, as shown in Fig 5. Overall, the NDWI shows a trend of periodic variation.

**Calculate BSI.** BSI demonstrates a strong enhancement effect for built-up bare soil areas. It not only enhances the degree and completeness of built-up areas but also effectively extracts some building areas that are not easily identifiable by other indices. The expression for BSI is as follows:

$$BSI = \frac{(SWIR + Red) - (SWIR + Blue)}{(SWIR + Red) + (SWIR + Blue)}$$

(5)

where Blue corresponds to band 2 in Landsat 8, representing the reflectance of the blue band.

The time-series BSI indices of the study area over the past decade have been calculated through the GEE platform, as shown in Fig 6. Overall, the BSI shows a trend of periodic variation.

## CA-Markov predict LUC

The CA-Markov model can consider changes in both space and time. Complex systems can have their spatial variations simulated by the CA model, while the Markov model excels in long-term prediction. The coupled CA-Markov model can perform long-term predictions effectively and simulate spatial changes efficiently.

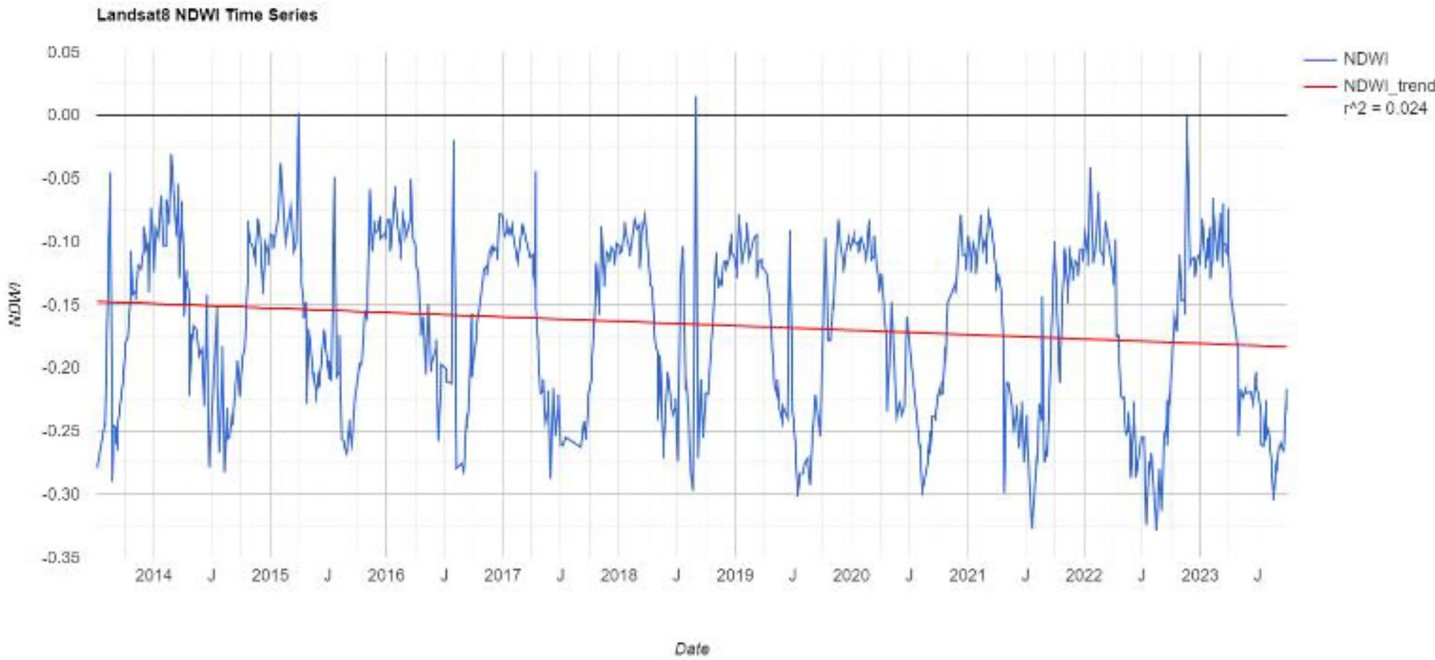

**Fig 5. The time series information of the NDWI.**

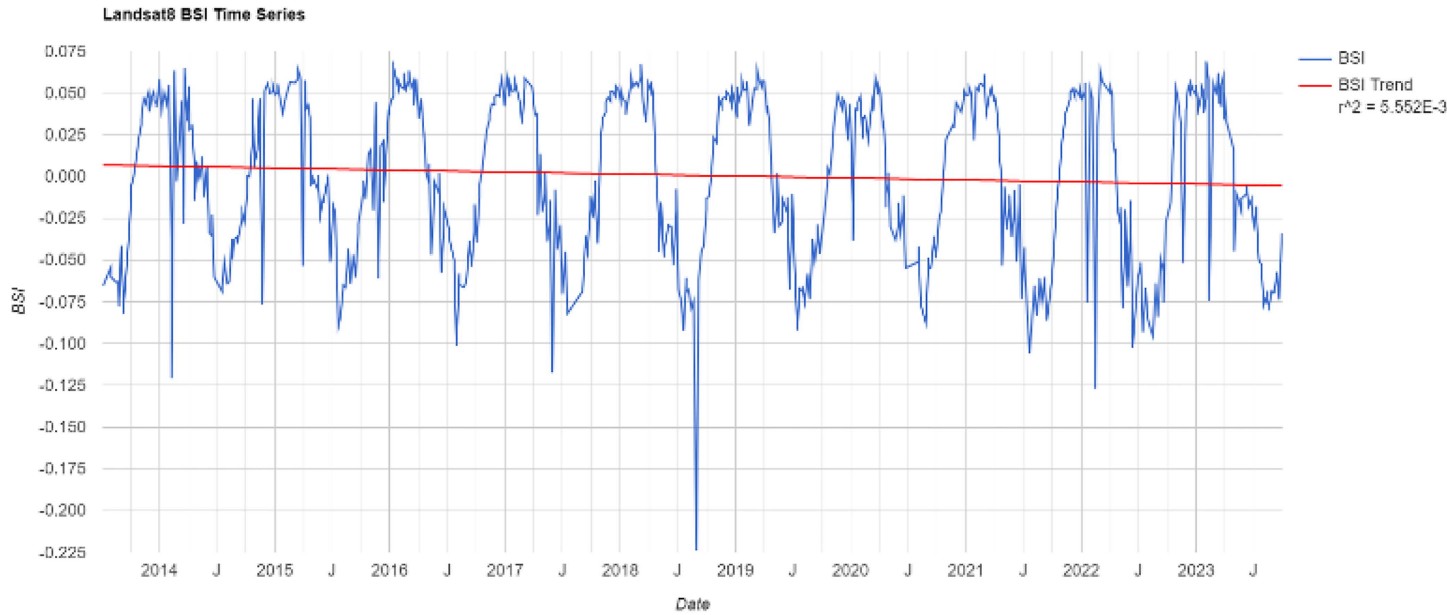

**Fig 6. The time series information of the BSI.**

A Markov chain is a stochastic process where a random variable transitions from one state to another. According to the properties of Markov chains, a variable's future state is determined by its present state alone. In a Markov chain, the transition probability from state i to state j determines the likelihood of this step in state transition. In the CA-Markov model, a pre-prepared Markov chain matrix is used. The states of cells are updated according to the states of adjacent cells, and the temporal changes are controlled by the transition probability matrix.

$$P(i \to j) = p\left[X = j \middle| X - 1 = i\right] \tag{6}$$

This step aims to obtain the land use results map for 2025, using land use maps from 2016, 2019, and 2022. Initially, a Markov chain is employed to calculate the probability transition matrix using data from 2016 and 2019. Subsequently, utilizing the data from 2019 as a baseline, the probability transition matrix obtained is used to simulate the land use result for 2022. The simulated result is iterated multiple times and compared with the actual land use data from 2022. Upon reaching near-saturation accuracy, the data from 2022 is then used as a baseline to forecast the land use data for 2025. The model diagram is as shown in Fig 7.

### Establish CNN-LSTM-attention model

This study predicts the temporal remote sensing indices of the study area by constructing a CNN-LSTM-Attention model, integrating remote sensing spectral indices calculated for the study area in 2007, 2010, 2013, 2016, 2019, and 2022.

LSTM neural networks are a special type of RNN model. Their unique architecture allows them to mitigate the vanishing gradient problem, enabling them to retain information from earlier time steps without incurring significant costs. LSTM consists of three main components, each responsible for a distinct function. Which details from the prior timestamp should be recalled or forgotten is decided in the first section. The second component, referred to as the cell, makes an effort to extract fresh data from the input. In the third section, the changed data is finally transmitted from the present timestamp to

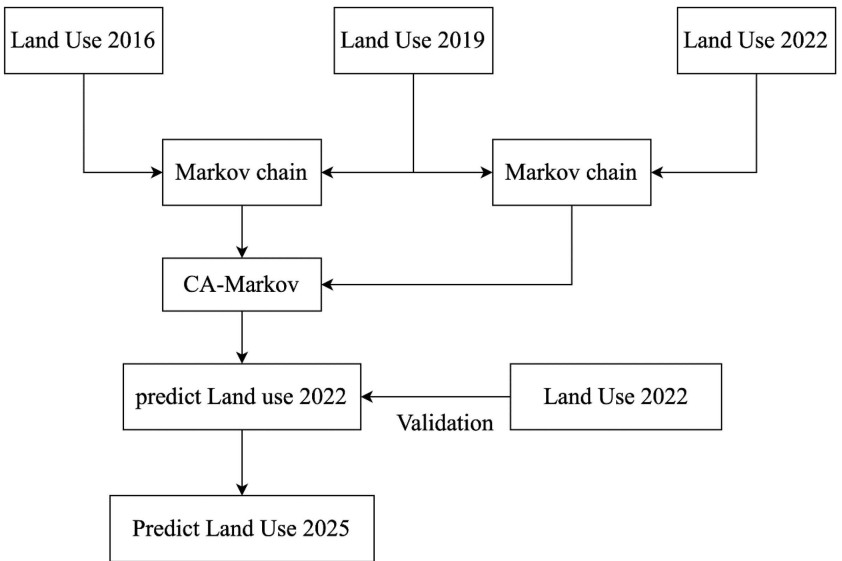

**Fig 7. CA-Markov model diagram.**

the subsequent timestamp.. We refer to these three parts of the LSTM unit as gates. The forget gate is the first, the input gate is the second component, and the third part is the output gate.

In order to improve the accuracy of the model under the condition of limited resources, it is necessary to allocate more resources to the key features of the objects to be recognized, that is, to enhance the recognition accuracy by means of weights. In this study, a channel attention mechanism is planned to be introduced to improve the performance of the model. The schematic diagram of the channel attention mechanism is shown in Fig 8.

The operation process of the SE (Squeeze-and-Excitation) channel attention mechanism is as follows: First, global average pooling is performed on the input features, and then they are fed into two fully connected layers. Among them, the first fully connected layer compresses the C channels into C/r channels to reduce the amount of calculation (here, r represents the compression ratio), and then it goes through a Relu non-linear activation layer. The number of neurons in the second fully connected layer is the same as that of the input feature layer, aiming to restore the number of channels to C. Finally, the Sigmoid function is used to limit the output value within the range of (0–1). This output value serves as the weight generated by the channel attention mechanism and is multiplied by the original features, so as to obtain the final features extracted after incorporating the attention mechanism.

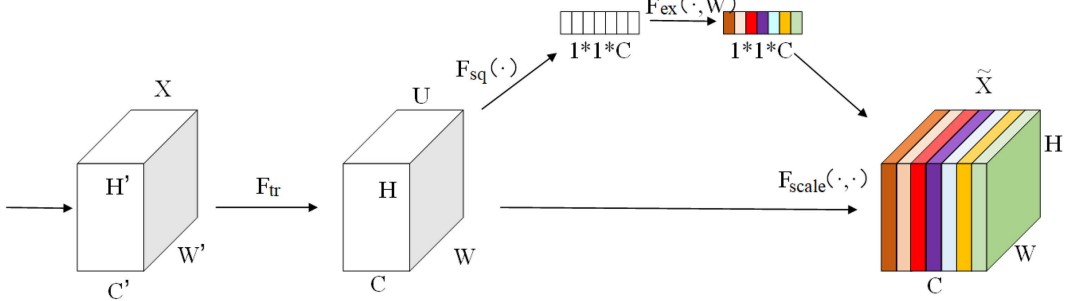

**Fig 8. Schematic diagram of the channel attention mechanism.**

The integrated model consists of an input layer, a Convolutional Neural Network (CNN) layer, a Long Short-Term Memory (LSTM) layer, an attention layer, and an output layer. In the input layer, the format of the input data is specified, with the batch size defaulting to 1 and the time step set to t = 3. The CNN layer can extract the spatial correlations among different feature values in the data. Within the CNN layer, convolution, pooling, and flattening operations are carried out sequentially. For the time-series data in this study, one-dimensional convolution is employed in the model. The LSTM layer is formed by stacking five LSTM layers. Dropout layers are added after the CNN layer and at the end of the LSTM layer to randomly discard nodes and prevent overfitting. The attention layer calculates the weighted sum of the LSTM output vectors. The vectors output by the LSTM layer serve as the input to the attention layer, which is then trained through a fully connected layer, followed by normalization of the output from the fully connected layer. The final output layer specifies the prediction time step and ultimately outputs the prediction results for the specified step. The schematic diagram of the CNN-LSTM-Attention model is shown in Fig 9.

To assess the predictive ability of the forecasting model, the dataset is randomly split divided into a training set and a test set, with 80% of the data used for training and the remaining for testing. Three metrics are employed in this study as error evaluation indicators: Mean Squared Error (MSE), Root Mean Squared Error (RMSE), and Coefficient of Determination ($R^2$).

$$MSE = \frac{1}{m} \sum_{i=1}^{m} (y_i - \hat{y}_i)^2 \tag{7}$$

$$RMSE = \sqrt{\frac{1}{m} \sum_{i=1}^{m} (y_i - \hat{y}_i)^2} \tag{8}$$

$$R^2 = 1 - \frac{\sum_{i=1}^{m}(y_i - \hat{y}_i)^2}{\sum_{i=1}^{m}(y_i - \overline{y}_i)^2} \tag{9}$$

where $m$ represents the number of samples, $y_i$ represents the true values, $\hat{y}_i$ represents the predicted values, and $\overline{y}_i$ represents the sample mean.

## Multiple linear regression model

Regression analysis is an important statistical method for analyzing the relationship between independent variables and dependent variables. It helps to determine the degree to which the dependent variable changes with the change in independent variables. The MLR model considers the effect of multiple independent variables on the dependent variable.

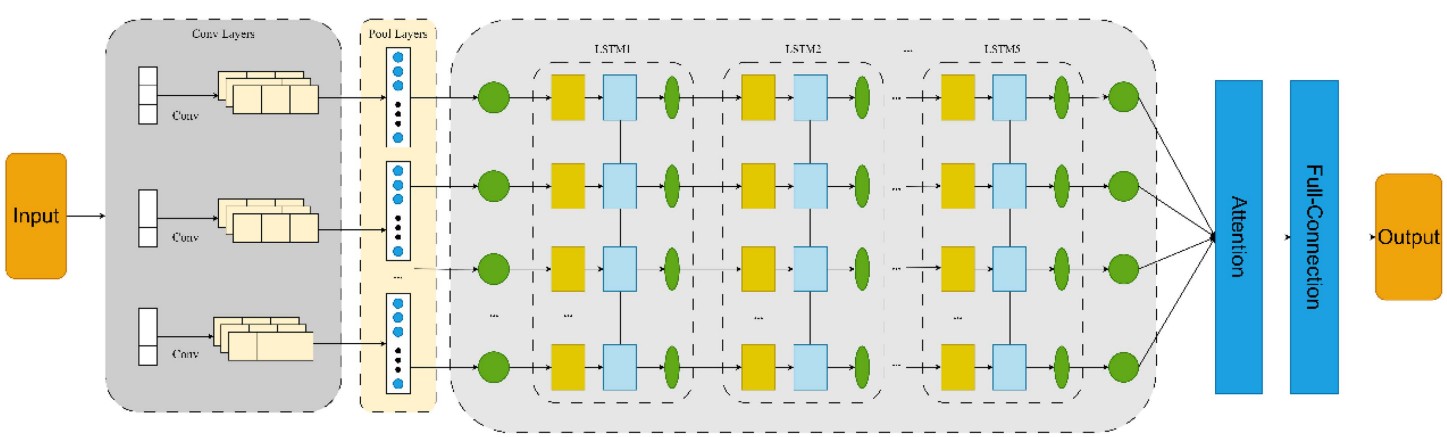

**Fig 9. CNN-LSTM-attention model diagram.**

$$y = a_0 + a_1 x_1 + a_2 x_2 + \ldots + a_n x_n \tag{10}$$

where $a_0$ is the intercept term, $a_0$, $a_1$, $a_2$,…, $a_n$ coefficient before the independent variable, representing weight; $x$ is the independent variable; and $y$ is the dependent variable.

Regression is widely used in many fields. To more accurately assess the relationship between UHII and remote sensing indices, corresponding land use categories must be considered when calculating input variables. For instance, when extracting NDVI values for vegetation categories, other areas in the region could be set to 0. As shown in Fig 10.

The accuracy of the regression model is evaluated through MSE, RMSE, and $R^2$.

## Results

### Land use change status in the study area

Fig 11 illustrates the land use change situation in the study area from 2013 to 2022.(Landsat 8).

Built-up areas account for the largest proportion of all land use categories and continue to expand, increasing from 2693.11 km² in 2013 to 2929.64 km² in 2022, an increase of 236.53 km². The proportion of bare land is gradually decreasing, reduced by 247.83 km². The changes in the proportions of vegetation and water bodies are not significant. The specific details are as shown in Table 1.

### Analysis of spatiotemporal variations in land surface temperature in the study area

Fig 12 illustrates the spatiotemporal patterns of LST in 2013, 2016, 2019, and 2022.(Landsat 8) To facilitate a visual comparison of LSTs on different dates, the following temperatures have been normalized. In the summer of 2013, LSTs ranged from 18.65°C to 57.86°C, and the average temperature is 34.61°C. LSTs ranged from 11.68°C to 58.62°C during the summer 2016, and the average temperature is 35.76°C. LSTs between 14.52°C to 54.18°C during the summer 2019, and the average temperature is 34.60°C. LSTs between 21.36°C to 57.78°C during the summer 2022, and the average temperature is 34.49°C.

In 2013, some areas within the central six districts exhibited relatively high LSTs. Additionally, there were several independent high-temperature regions located in Shunyi District, Changping District, and Shijingshan District. From 2013

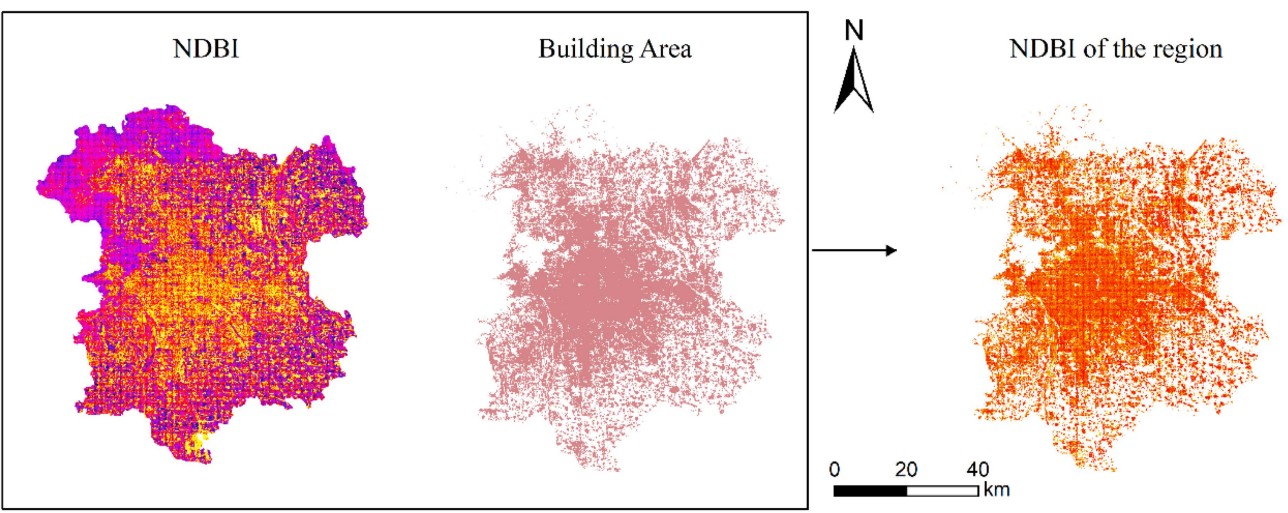

**Fig 10. The NDBI index extracted based on land use categories.** (Landsat 8).

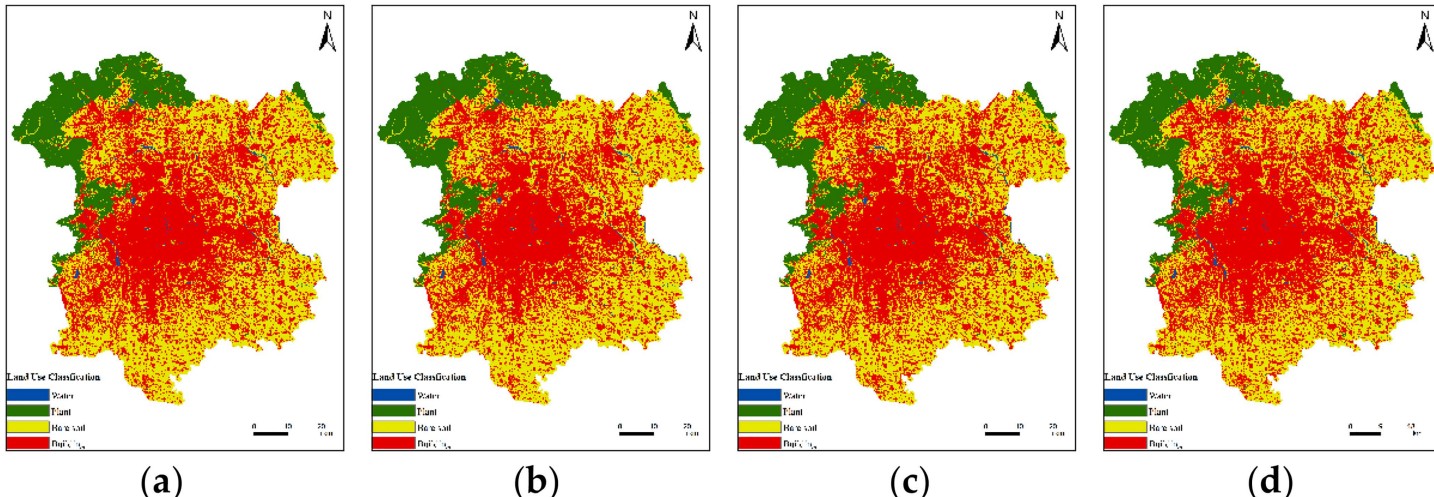

**Fig 11. The four images above correspond to LUC in 2013, 2016, 2019, and 2022 respectively. (a) LUC in 2013; (b) LUC in 2016; (c) LUC in 2019; (d) LUC in 2022.**

**Table 1. Land use change from 2013 to 2022.**

| Land use classification | Area and percentage | 2013 | 2016 | 2019 | 2022 |
|---|---|---|---|---|---|
| **Water** | Area (km$^2$) | 50.51 | 49.49 | 48.26 | 53.00 |
| | Percentage (%) | 0.81 | 0.79 | 0.77 | 0.85 |
| **Plant** | Area (km$^2$) | 883.54 | 887.25 | 887.43 | 892.35 |
| | Percentage (%) | 14.13 | 14.19 | 14.19 | 14.27 |
| **Bare soil** | Area (km$^2$) | 2626.14 | 2518.48 | 2451.61 | 2378.31 |
| | Percentage (%) | 41.99 | 40.27 | 39.21 | 38.03 |
| **Building** | Area (km$^2$) | 2693.11 | 2798.08 | 2866.00 | 2929.64 |
| | Percentage (%) | 43.07 | 44.75 | 45.83 | 46.85 |

and 2016, the high-temperature areas in the central six districts began to expand, gradually connecting the originally scattered high-temperature patches in the surrounding areas. From 2016 to 2019, the high-temperature areas further expanded, covering more than half of the study area, and there was a trend of overall movement towards the southwest. High-temperature patches began to appear in Tongzhou District and Daxing District. Between 2019 and 2022, the average temperature in the entire study area remained almost unchanged compared to 2019, but the temperature distribution became more uniform. The central six districts still maintained relatively high LSTs, and the area northeast of Chaoyang District experienced further temperature increases. The average maximum and minimum temperatures in the study area both show a trend of further increase. Fig 13 is a line graph depicting the surface temperature changes in the study area from 2013 to 2022.

## Analysis of UHI evolution characteristics

The calculated UHII levels are first normalized to eliminate the effects of different times and weather conditions on the acquired images, facilitating subsequent analysis. Then, the normalized data is categorized. Common methods for categorization include equal interval method and mean-standard deviation method. Studies have shown that the

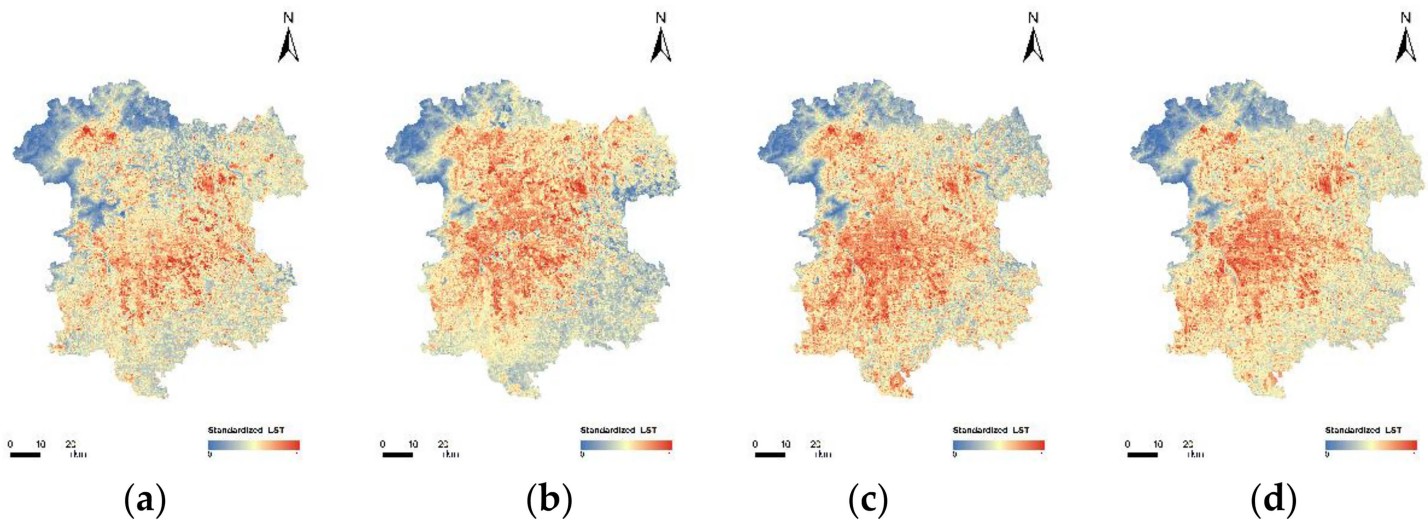

**Fig 12. Maps of LST distribution in the study area for the years 2013, 2016, 2019, and 2022. (a) LST in 2013; (b) LST in 2016; (c) LST in 2019; (d) LST in 2022.**

mean-standard deviation method is more in line with the actual situation of the research area. This study uses the mean-standard deviation method to classify UHII.

Fig 14 illustrates the categorization of UHII levels using the mean-standard deviation method.(Landsat 8) The intensity levels include: Strong Cold Island Area, Cold Island Area, Weak Heat Island Area, Heat Island Area, and Strong Heat Island Area. The criteria for categorization are shown in Table 2.

Although the acquisition times of the images and environmental factors vary, the UHI effect is evident in each of the years. From 2013 to 2022, the UHI effect in the six central districts has been strong. These areas are characterized by heavy traffic, high population density, numerous commercial centers, and are the economic development hubs. With the steady progress of urbanization, the areas with strong heat islands have gradually expanded, connecting many originally fragmented strong heat island patches and mainly moving southwestward. The regions with abundant vegetation and forests in the northwest, as well as areas with river distributions, are the low-temperature zones during all periods.

From Table 3, we can compare the UHII of the research area in 2013, 2016, 2019, and 2022. The results show that the total area of heat island and strong heat island regions increased by 39.97 square kilometers. However, the heat island situation in 2022 has improved compared to 2019. Over these ten years, the weak heat island regions have predominantly constituted the UHI. The proportion of the study area with a weak heat island effect decreased from 39.29% in 2013 to 35.37% in 2016, but then rose again to 39.22% in 2019, matching the 2013 proportion. The overall proportion of heat island and strong heat island regions increased year by year, with a slight decrease from 2019 to 2022. The 2022 Winter Olympics, hosted by Beijing, makes Beijing the first "Dual Olympic City" in the world. During the preparation period, a series of environmental improvement measures in Beijing effectively alleviated the overall UHI effect. The comparison of UHII is shown in Fig 15.

### Predicted land use distribution for 2025 using the CA-Markov model

To determine the accuracy of the developed CA-Markov model, a comparison between the actual LUC map in 2022 and the simulation results was conducted. As shown in Table 4. The Kappa coefficient is 71.59%, indicating overall good performance. The predicted 2025 LUC map is presented in Fig 16. It is anticipated that the built-up area will continue to

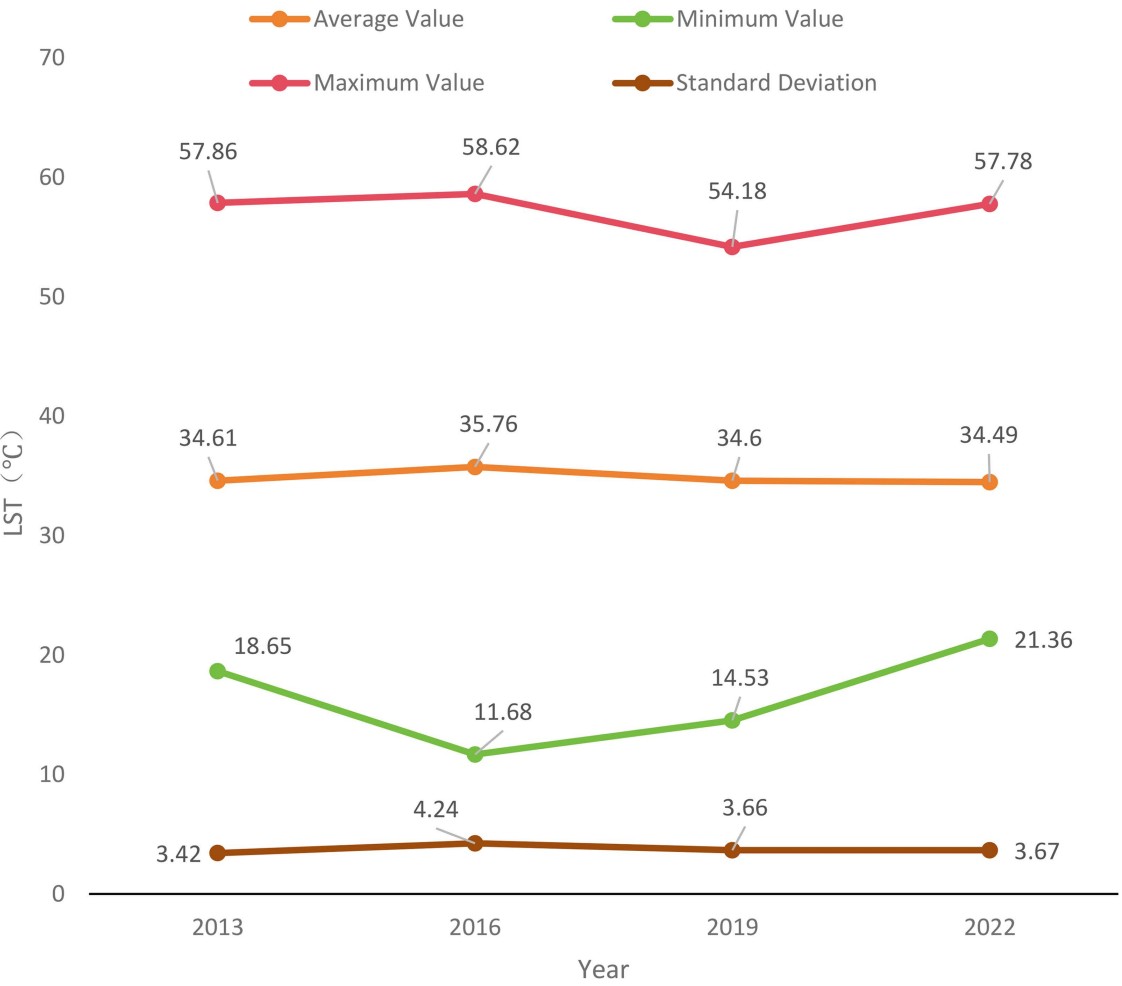

**Fig 13. Changes in LST from 2013 to 2022.**

grow by 2025, constituting nearly 50% of the total study region. The continuous expansion of the built-up area will connect initially scattered building patches into complete, continuous blocks. Bare land area is expected to shrink further, with most bare soil areas acting as the main source of expansion for built-up areas. Compared to 2022, the bare soil area is predicted to decrease by 74.92 km², while the built-up area is expected to increase by 85.75 km².

### Predicted results of the CNN-LSTM-attention model

Train the CNN-LSTM-Attention model by changing its hyperparameters, including the structure, the number of epochs, the learning rate, the activation function, etc. When the results reach a satisfactory accuracy, determine the corresponding weight coefficients. Since the model conducts time series prediction for individual remote sensing indices, they have similar hyperparameters. Fig 18 below shows the distribution map of the remote sensing indices in 2022 simulated based on the remote sensing indices of historical years, and it is compared with the actual distribution of the remote sensing indices in 2022. Fig 17 is a scatter plot of the errors of the simulated distribution of the remote sensing indices. The error information of the four predicted remote sensing indices is shown in Table 5.

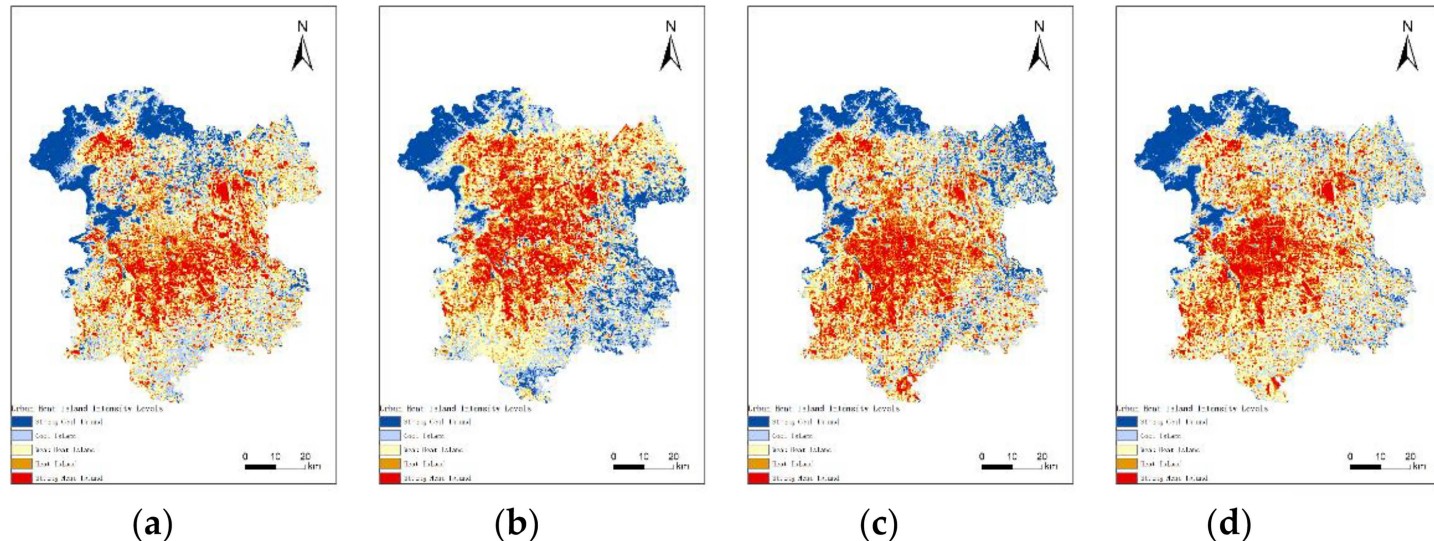

(a) (b) (c) (d)

**Fig 14. UHII in 2013, 2016, 2019, and 2022.** (a) UHII in 2013; (b) UHII in 2016; (c) UHII in 2019; (d) UHII in 2022.

**Table 2. Basis for classification of UHII.**

| Heat island zoning | Heat island class | Ecological assessment |
|---|---|---|
| N<mean-std | Strong cool island | Excellent |
| mean-std<N<mean-0.5std | Cool island | Favorable |
| mean-0.5std<N<mean+0.5std | Weak heat island | General |
| mean+0.5std<N<mean+std | Heat island | Mediocre |
| N> mean+std | Strong heat island | Poor |

**Table 3. Area statistics for UHII classification.**

| Heat Island Class | Area and percentage | 2013 | 2016 | 2019 | 2022 |
|---|---|---|---|---|---|
| **Strong Cool Island** | Area (km²) | 888.94 | 938.61 | 1019.58 | 839.39 |
| | Percentage (%) | 14.21 | 15.01 | 16.30 | 13.42 |
| **Cool Island** | Area (km²) | 1051.89 | 1032.11 | 965.90 | 1063.38 |
| | Percentage (%) | 16.82 | 16.51 | 15.45 | 17.01 |
| **Weak Heat Island** | Area (km²) | 2456.67 | 2361.76 | 2211.95 | 2452.76 |
| | Percentage (%) | 39.29 | 37.76 | 35.37 | 39.22 |
| **Heat Island** | Area (km²) | 886.37 | 879.79 | 999.25 | 894.25 |
| | Percentage (%) | 14.17 | 14.07 | 15.98 | 14.30 |
| **Strong Heat Island** | Area (km²) | 969.43 | 1041.02 | 1056.63 | 1003.52 |
| | Percentage (%) | 15.51 | 16.65 | 16.90 | 16.05 |

Based on the validation results showing satisfactory prediction accuracy, the historical remote sensing index data for the study area in the years 2007, 2010, 2013, 2016, 2019, and 2022 will be used to predict the distribution of remote sensing indices for the year 2025, as required by this study. As shown in Fig 19.(Landsat 8).

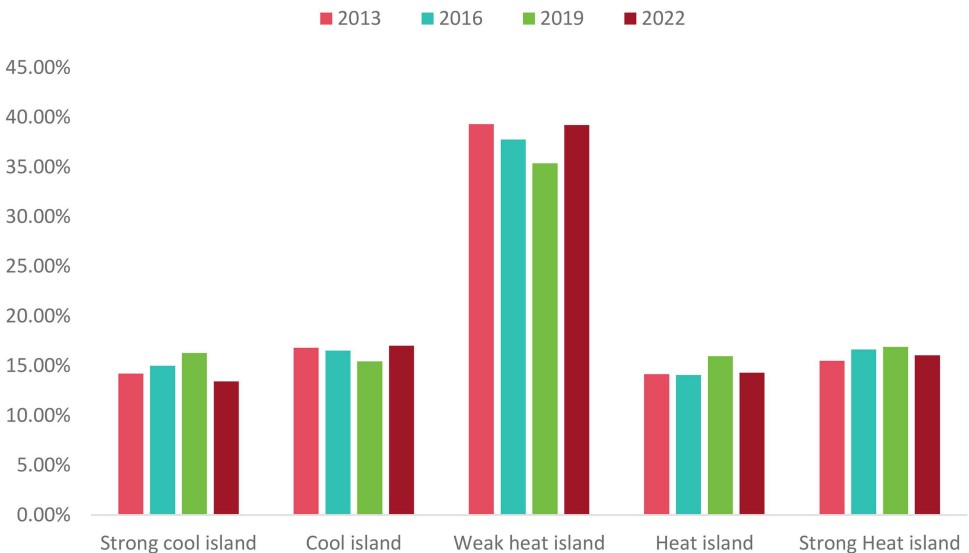

**Fig 15. Comparison of UHII in 2013, 2016, 2019, and 2022.**

**Table 4. Comparative analysis of LUC distribution between 2025 and 2022.**

| Land use classification | Area and percentage | 2022 | 2025 |
|---|---|---|---|
| **Water** | Area (km²) | 53.00 | 50.68 |
| | Percentage (%) | 0.85 | 0.81 |
| **Plant** | Area (km²) | 892.35 | 883.84 |
| | Percentage (%) | 14.27 | 14.13 |
| **Bare soil** | Area (km²) | 2378.31 | 2303.39 |
| | Percentage (%) | 38.03 | 36.84 |
| **Building** | Area (km²) | 2929.64 | 3015.39 |
| | Percentage (%) | 46.85 | 48.22 |

## Results of multivariate linear regression model

Due to the different ranges of variation between UHII and remote sensing indices, all parameters were normalized, and multivariate linear regression was performed on the normalized results. This effectively avoids bias towards indices with higher numerical values. After establishing the regression relationship and determining satisfactory accuracy, the results were analyzed. UHII is negatively correlated with NDVI and NDWI, indicating that UHII decreases with the increase of vegetation areas and water bodies. UHII was positively correlated with NDBI and BSI, suggesting that UHII increases with an increase in impervious surfaces and barren land. Additionally, the analysis revealed that as one of the economic and cultural centers in China, Beijing's developed urban construction and high-density building clusters contribute to stronger UHI effects, especially in densely built-up areas. The impact of building indices on UHII in the study area is the greatest, followed by bare soil areas and vegetation areas, while water bodies have the least impact. The UHI effect is weaker in areas with more water bodies. Therefore, water bodies can mitigate UHI effects to some extent in the study area.

$$UHII = 0.33 + 0.35 * NDBI - 0.09 * NDVI - 0.04 * NDWI + 0.17 * BSI \tag{11}$$

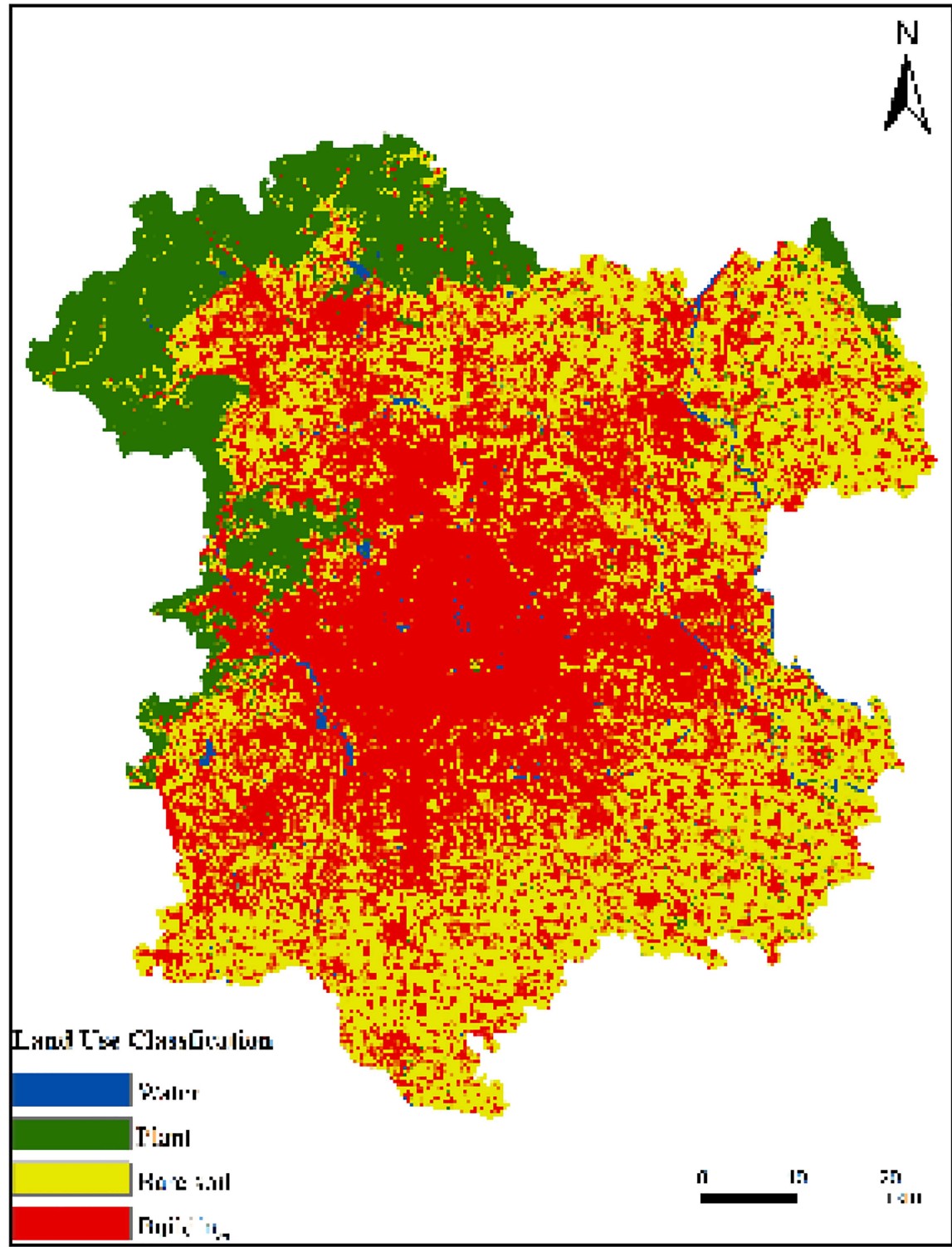

**Fig 16.  Predicted 2025 LUC Map. (Landsat 8).**

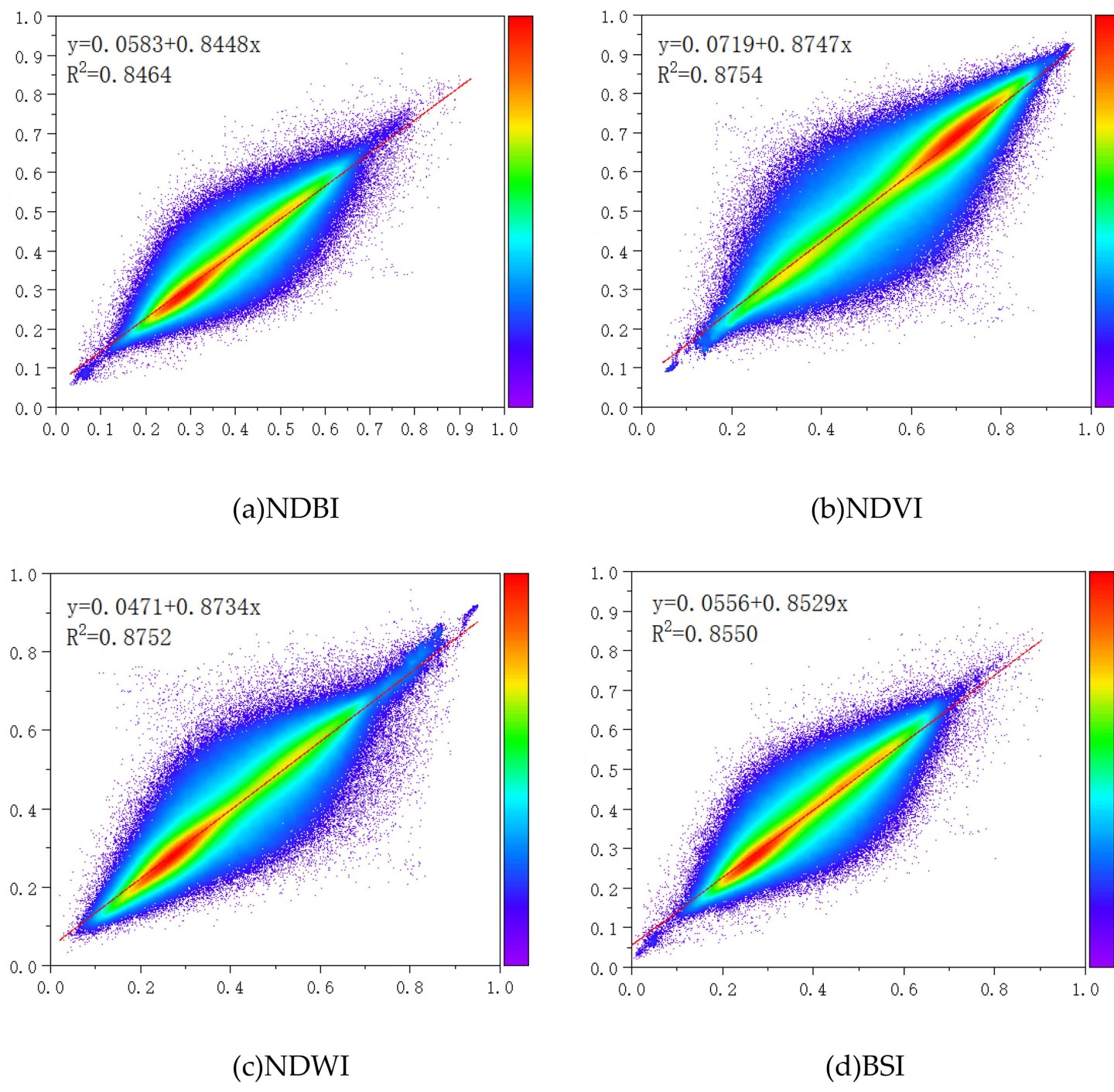

**Fig 17. The scatter plot of the errors of the simulated distribution of the remote sensing indices.** (a) NDBI (b) NDVI (c) NDWI (d) BSI.

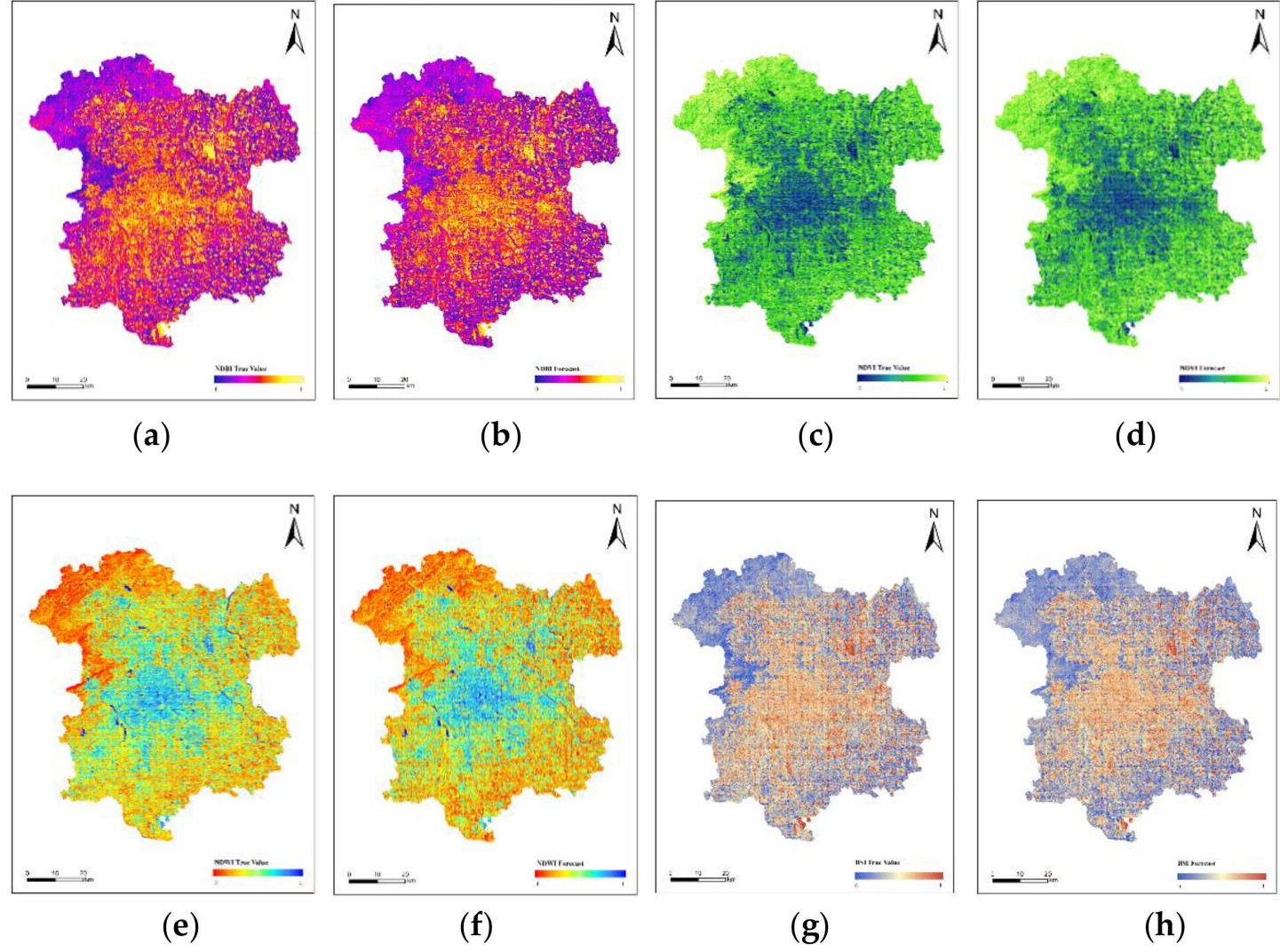

**Fig 18. Comparison between the predicted and actual values of NDBI, NDVI, NDWI, and BSI for the year 2022.** (a) The actual values of NDBI for the year 2022; (b) The predicted values of NDBI for the year 2022; (c) The actual values of NDVI for the year 2022; (d) The predicted values of NDVI for the year 2022; (e) The actual values of NDWI for the year 2022; (f) The predicted values of NDWI for the year 2022; (g) The actual values of BSI for the year 2022; (h) The predicted values of BSI for the year 2022.

**Table 5. Error information for the predicted four remote sensing indices.**

| Remote sensing spectral indices | $R^2$ | RMSE | MSE |
|---|---|---|---|
| NDBI | 0.8464 | 0.0423 | 0.0017 |
| NDVI | 0.8754 | 0.0579 | 0.0033 |
| NDWI | 0.8752 | 0.0496 | 0.0024 |
| BSI | 0.8550 | 0.0450 | 0.0020 |

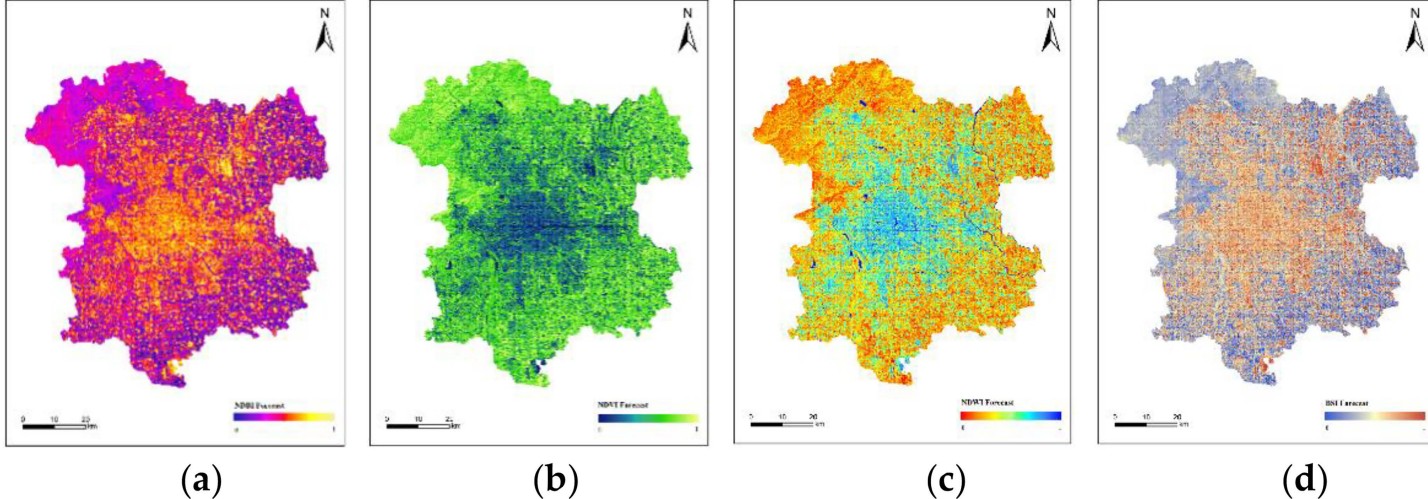

**Fig 19. Predicted values of NDBI, NDVI, NDWI, and BSI for the year 2025. (a) Predicted NDBI Values for 2025; (b) Predicted NDVI Values for 2025; (c) Predicted NDWI Values for 2025; (d) Predicted BSI Values for 2025.**

The extracted remote sensing indices from 2022 were input into the constructed regression model. The model's simulated findings and the real UHII outcomes from 2022 were contrasted., and the errors are shown in Table 6.

The remote sensing index results predicted by the CNN-LST-Attention model were input into the constructed MLR model to forecast the UHII for the year 2025(Fig 20). Classification of the prediction results and comparison with the results in 2022, analysis of the distribution and variation of UHII. Table 7 shows the statistical results of the areas for each level of UHII in 2025, while Fig 21 illustrates the comparison of UHII distribution between 2025 and 2022. The proportion of cold island areas has decreased significantly, and the proportion of strong heat island areas has slightly decreased. Despite this, the overall extent of the heat island has expanded. In comparison to 2022, there was a 2.97% increase in the proportion of the heat island area, and an additional 1.25% rise in the combined areas of the heat island and strong heat island. These findings suggest that the intensity of the heat island effect within the study region has escalated.

## Discussion

Significant changes have been made in the LUC in Beijing over the past decade, primarily due to the replacement of bare land areas with developed areas. This transformation is a result of economic development. The "Beijing Urban Master Plan (2016-2035)", issued by the Beijing Municipal Committee and Municipal Government in 2017, focuses on planning the future of the capital with a broader spatial perspective. It adheres to the rigid constraints of resource and environmental carrying capacity, sets limits on population size, establishes ecological control lines and urban development boundaries, and shifts from expansive planning to optimizing spatial structure as a development strategy (Beijing Municipal Government, 2017), which will have profound implications for Beijing's land use pattern. In recent years, Beijing has continuously hosted large international conferences, coupled with the successful hosting of the Beijing Winter Olympics in 2022, further increasing the demand for construction land in Beijing. The continuous expansion of developed areas has

**Table 6. Analysis of the discrepancy between actual and simulated values of UHII in 2022.**

|  | R² | RMSE | MSE |
|---|---|---|---|
| **UHII** | 0.7468 | 0.0546 | 0.0029 |

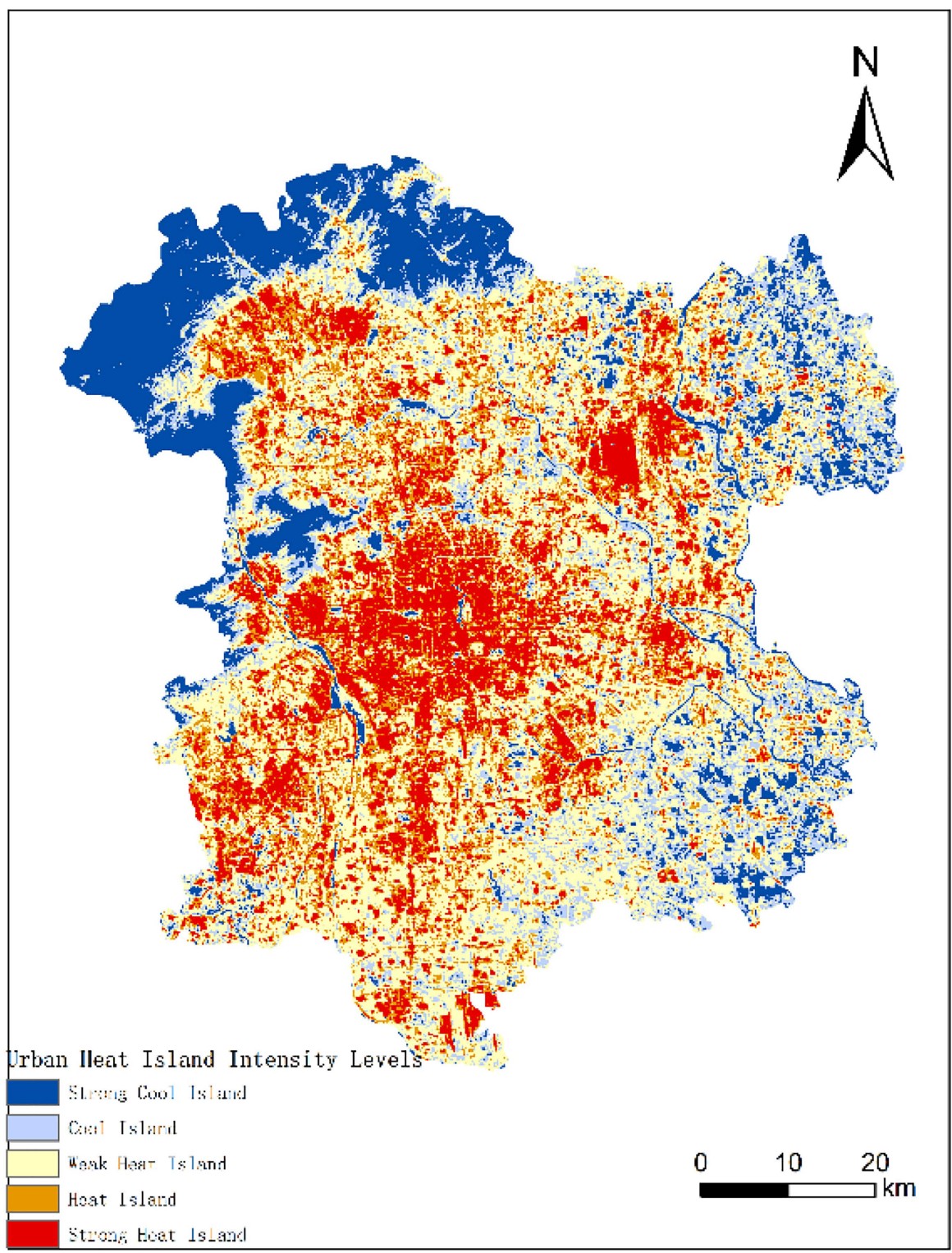

**Fig 20. UHII and its distribution in 2025. (Landsat 8).**

**Table 7.** Comparison and analysis of UHII between 2025 and 2022.

| Heat island class | Area and percentage | 2022 | 2025 |
|---|---|---|---|
| **Strong Cool Island** | Area (km²) | 839.39 | 932.06 |
| | Percentage (%) | 13.42 | 14.90 |
| **Cool Island** | Area (km²) | 1063.38 | 785.23 |
| | Percentage (%) | 17.01 | 12.56 |
| **Weak Heat Island** | Area (km²) | 2452.76 | 2560.13 |
| | Percentage (%) | 39.22 | 40.94 |
| **Heat Island** | Area (km²) | 894.25 | 978.09 |
| | Percentage (%) | 14.30 | 15.64 |
| **Strong Heat Island** | Area (km²) | 1003.52 | 997.79 |
| | Percentage (%) | 16.05 | 15.96 |

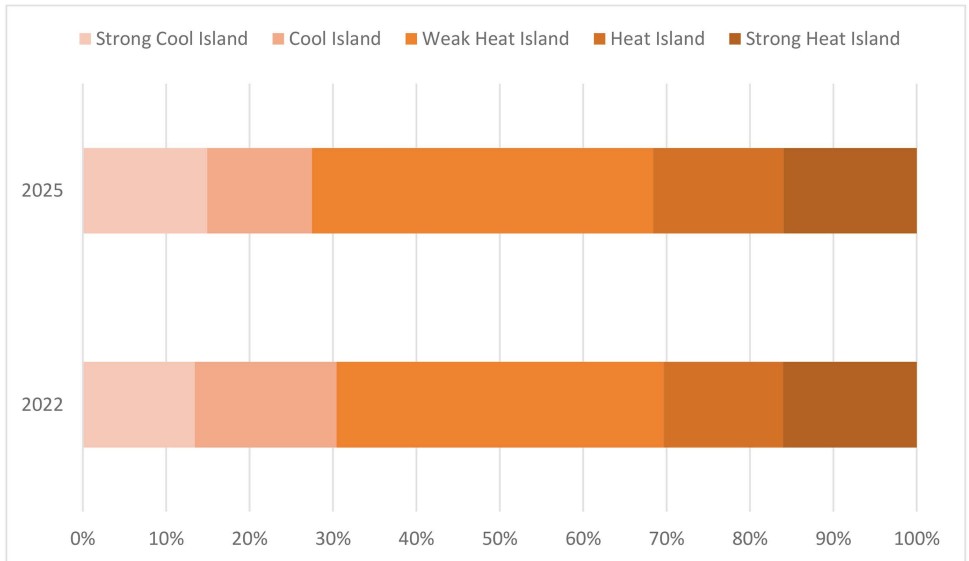

**Fig 21. Comparison of UHII between 2025 and 2022.**

led to an increase in UHIs. The concentration of high-temperature areas in developed areas can be observed by comparing the distribution maps of LST and LUC.

Selecting the appropriate method/model is crucial for accurately predicting future UHII. The use of MLR models can effectively identify and differentiate the key variables influencing UHII. By determining the relationship between UHII and the key variables, the MLR model can be applied for prediction. However, regression models have inherent limitations. Solely focusing on prediction accuracy, deep learning or artificial neural networks can handle nonlinear relationships, improving prediction accuracy with higher R2 and lower RMSE. However, both artificial neural networks and deep learning function as "black boxes," making it challenging to identify the precise causes of severe UHII. Therefore, regression models should be used in order to identify the primary affecting factors. Subsequently, optimization of the UHII prediction model can be achieved by combining deep learning and artificial neural networks.

According to the predicted LUC map, the developed areas will continue to expand in various directions, leading to a significant increase in impervious surfaces. This expansion will result in the absorption of more energy in the region, increasing UHII by continually raising the surface temperature of the surrounding areas. This situation may lead to a sharp decline in the livability of Beijing, which already has a dense population, with shortages in living spaces and decreased utilization of natural resources.

The coordinated development of Beijing-Tianjin-Hebei effectively alleviates the pressure on Beijing's non-capital functional areas. Xiong'an New Area, as a millennium plan in China, plays a key role in promoting the coordinated development of Beijing-Tianjin-Hebei. By adhering to and optimizing the core functions of the capital, as well as adjusting and weakening functions that are not suitable for the capital, a series of measures implemented and enforced will effectively enhance the livability of Beijing.

Currently, almost all major cities globally are facing rising temperatures and increasing UHI effects. Therefore, urban cooling initiatives are considered activities directly related to economic growth and inseparable from sustainable development goals. Effective urban cooling goals can be achieved by issuing a series of policies and regulations. Research analysis reveals that LST near vegetation and water bodies is lower, resulting in weaker UHI effects. It's possible to alleviate the UHI effect by reducing the area of developed land and increasing the area covered by vegetation areas and water bodies. However, in the current scenario of rapid urbanization, significantly reducing the area of developed land is impractical. Still, controlling the urban development boundary can regulate the spread of developed areas. Based on the analysis of the regression results, we can make the following recommendations for improving urban livability and urban sustainability: (1) Reduce the planned area of developed land. (2) Decrease the density of bare land areas and increase the proportion of vegetation areas and water body coverage.

## Conclusions

The objective of this study is to examine the distribution and evolution trend of UHII in relation to land use classification projected for 2025. By constructing a CNN-LSTM-Attention model, predictions were made for four remote sensing spectral indices: NDBI ($R^2 = 0.8464$), NDVI ($R^2 = 0.8754$), NDWI ($R^2 = 0.8752$), and BSI ($R^2 = 0.8550$). Subsequently, a MLR model was built to predict UHII ($R^2 = 0.7468$, RMSE $= 0.0546$). The model demonstrates good accuracy, aiding in accurate UHII predictions. Due to the further expansion of urbanized areas, the UHI effect in Beijing will intensify. To mitigate the risks to urban livability and sustainable development, this study recommends controlling the urban development boundary, increasing the proportion of vegetation areas and water body areas, and optimizing the urban development pattern.

In conclusion, this study provides a method for predicting future UHII by combining MLR model with deep learning. Future UHII forecasts were achieved by the study using CA-Markov to predict future land use classification results. Using a MLR model to support direct urban development planning recommendations was proposed, with the goal of improving urban livability and achieving sustainable development.

## Author contributions

**Conceptualization:** Xinran Liu, xia zhu, Cui Jia.

**Data curation:** Jie Cao, Yu Zhong.

**Formal analysis:** Xinran Liu, Cui Jia, Jie Cao.

**Investigation:** Yu Zhang, Yunjie Zhang.

**Methodology:** Xinran Liu, xia zhu, Cui Jia.

**Project administration:** xia zhu.

**Resources:** Xinran Liu, Jie Cao, Yu Zhong, Yu Zhang, Yunjie Zhang.

**Software:** Yu Zhong.

**Supervision:** xia zhu, Yuanping Liu.

**Validation:** Xinran Liu, Jie Cao, Yu Zhong, Yu Zhang, Yunjie Zhang.

**Writing – original draft:** Xinran Liu, Yuanping Liu, Yunjie Zhang.

**Writing – review & editing:** Xinran Liu, xia zhu, Yuanping Liu.

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
