## [Decision Letter · Decision Letter 0]

2 May 2025

Dear Dr. zhu,

We look forward to receiving your revised manuscript.

Kind regards,

Lingye Yao, Ph.D.

Academic Editor

PLOS ONE

3. We note that Figures 1,10, 11, 12, 14, 16, 18, 19 and 20 in your submission contain [map/satellite] images which may be copyrighted. All PLOS content is published under the Creative Commons Attribution License (CC BY 4.0), which means that the manuscript, images, and Supporting Information files will be freely available online, and any third party is permitted to access, download, copy, distribute, and use these materials in any way, even commercially, with proper attribution. For these reasons, we cannot publish previously copyrighted maps or satellite images created using proprietary data, such as Google software (Google Maps, Street View, and Earth). For more information, see our copyright guidelines: http://journals.plos.org/plosone/s/licenses-and-copyright.

a. You may seek permission from the original copyright holder of Figures 1,10, 11, 12, 14, 16, 18, 19 and 20 to publish the content specifically under the CC BY 4.0 license. 

Natural Earth (public domain): http://www.naturalearthdata.com

Reviewers' comments:

Reviewer's Responses to Questions

**Comments to the Author**

1. Is the manuscript technically sound, and do the data support the conclusions?

Reviewer #1: Yes

Reviewer #2: Yes

Reviewer #3: Yes

2. Has the statistical analysis been performed appropriately and rigorously?

Reviewer #1: Yes

Reviewer #2: Yes

Reviewer #3: I Don't Know

3. Have the authors made all data underlying the findings in their manuscript fully available?

Reviewer #1: Yes

Reviewer #2: Yes

Reviewer #3: Yes

4. Is the manuscript presented in an intelligible fashion and written in standard English?

Reviewer #1: Yes

Reviewer #2: Yes

Reviewer #3: Yes

Reviewer #1: This study combines CA-Markov model, CNN-LSTM-Attention deep learning model and multiple linear regression (MLR) to predict the spatial and temporal distribution of urban heat island intensity (UHII) in Beijing in 2025, which is of clear practical significance. The research design is relatively complete and the methodology is innovative, but some details need to be further optimised to enhance scientific rigour. Therefore, I suggest a minor revision.

1. The details of the calculation of the transfer probability matrix are not explicitly stated in the CA-Markov model Suggested additions. (e.g. whether based on statistical analyses of historical land-use change).

2. Only four spectral indices, NDBI, NDVI, NDWI, and BSI, were included in the MLR model, and other potential influences were not considered, and it is suggested to add or explain why. (e.g. population density, traffic flow, building density).

3. The CA-Markov model validation relied only on the Kappa coefficient (71.59%) and it is recommended that other metrics (e.g., FoM index) be supplemented to fully assess the simulation performance.

4. UHII classification criteria do not cite existing studies or local standards and require additional justification.

5. The decrease in the proportion of heat island area in 2022 compared to 2019 may be related to the environmental policies of the Winter Olympics, but this effect is not explicitly analysed in the discussion, and it is recommended that the analysis of the correlation between the policy factors and the results be supplemented..

6. The conclusion ‘Recommended Optimised Urban Development Patterns’ is rather general, and it is recommended that specific measures (e.g. which areas of vegetation or water bodies should be prioritised for protection) should be proposed in conjunction with the results of the projections.

7. Some of the references are not formatted uniformly, and it is recommended that they be adjusted according to the requirements of the journals.

Reviewer #2: 1. Integrating CA-Markov and artificial neural networks for UHII-related predictions is not new. The authors should better clarify the limitations of previous studies or explain why those approaches may be less suitable for the current context. More importantly, they need to highlight their model's distinct contributions. This should be discussed more clearly in both the introduction and discussion sections.

2. Although the authors provide R² and RMSE to evaluate the models, it is recommended to include additional indexes such as MAE and MAPE to better assess performance. Furthermore, the training process lacks information regarding model generalization and overfitting control. Including training and validation loss curves to demonstrate model convergence and stability.

3. In the multiple linear regression section, while the relationship between indices like NDVI and UHII is described, the statistical robustness of the regression is not fully demonstrated. The manuscript should report the p-values for regression coefficients and VIF values to test for significance and multicollinearity.

4. For the Landsat image processing, the authors must specify which spectral bands were used and clarify whether cloud masking and atmospheric correction were applied before further analysis. Regarding the UHII definition, it is essential to explain how the boundaries between urban and suburban areas were determined. Additionally, the manuscript should consistently use Celsius or Kelvin for temperature.

5. A confusion matrix or Kappa coefficient can provide a more accurate classification performance assessment. Since subsequent prediction results rely heavily on land use classification, its uncertainty and potential impact should be discussed.

6. In the spectral index computation (NDVI, NDBI, NDWI, BSI), the authors mention using the Google Earth Engine (GEE) but do not clarify the temporal range (e.g., annual mean, summer average, specific dates). The figures showing index time series (e.g., Fig. 4) should indicate the temporal resolution and include units.

7. The meaning of “time step t=3” should be clarified—does this refer to three consecutive periods? If so, is a sliding window input approach used?

8. There is redundancy between Table 3 and Figure 15, as both display similar information. One of them may be removed or merged for conciseness.

9. A table listing the complete CNN-LSTM-Attention model hyperparameters, such as batch size, learning rate, number of epochs, and optimizer settings.

10. Please add a glossary of technical terms.

Reviewer #3: dear Authors, assuming that all the technical-statistical parts are correct, I think your paper is already fine.

I only have a comment: what about adding in your introduction a little section (a few paragraphs) listing possible planning mitigation to urban heat island (e.g. Isobenefit Urbanism, or Green Wedge Urbanism)? I think it will provide an useful addition for the readers and the paper. Especially considering the new few billion urban dwellers expected in next decades, namely new cities to be built using such new urban approaches mitigating substantially urban heat island effects.

**Do you want your identity to be public for this peer review?** For information about this choice, including consent withdrawal, please see our Privacy Policy

Reviewer #1: No

Reviewer #2: No

Reviewer #3: No

---

## [Author Response · Author response to Decision Letter 1]

14 Oct 2025

The detailed content of the reply is in the attached letter

---

## [Decision Letter · Decision Letter 1]

21 Nov 2025

Dear Dr. zhu,

We look forward to receiving your revised manuscript.

Kind regards,

Lingye Yao, Ph.D.

Academic Editor

PLOS ONE

Journal Requirements:

Additional Editor Comments:

Please carefully address the minor review comments raised by Reviewer 3.

Reviewers' comments:

Reviewer's Responses to Questions

**Comments to the Author**

Reviewer #2: All comments have been addressed

Reviewer #3: (No Response)

2. Is the manuscript technically sound, and do the data support the conclusions?

Reviewer #2: Yes

Reviewer #3: Yes

3. Has the statistical analysis been performed appropriately and rigorously?

Reviewer #2: Yes

Reviewer #3: N/A

4. Have the authors made all data underlying the findings in their manuscript fully available?

Reviewer #2: Yes

Reviewer #3: Yes

5. Is the manuscript presented in an intelligible fashion and written in standard English?

Reviewer #2: Yes

Reviewer #3: Yes

Reviewer #2: The authors have fully addressed my previous comments, and the manuscript is now suitable for publication.

Reviewer #3: Dear Authors,

I guess you sufficiently replied to all the previous reviewers' comments. Even if they should be the ones claiming so.

Taking for granted the above, I only a minor comment: in your introduction, together with "Green Wedge Urbanism", and "Low-Carbon Urbanism", you could find interesting the "Isobenefit Urbanism" (IU) approach too regarding the UHI due to its constant ratio between green and built land regardless the settlement size (from village to megacity).

Congratulations again for your interesting article which I find ready.

**Do you want your identity to be public for this peer review?** For information about this choice, including consent withdrawal, please see our Privacy Policy

Reviewer #2: No

Reviewer #3: No

---

## [Author Response · Author response to Decision Letter 2]

9 Dec 2025

Thank you for this valuable suggestion. We agree that Isobenefit Urbanism (IU) offers a meaningful perspective. However, given the journal's strict length constraints for the Introduction section, we find it challenging to adequately elaborate on IU's theoretical dimensions without making this part overly fragmented. We have briefly touched upon the equity issues of cross-scale planning in the Discussion section, and we hope to dedicate a future paper to exploring how IU can be integrated with our research methodology in greater depth.

---

## [Decision Letter · Decision Letter 2]

11 Dec 2025

Prediction of Urban Heat Island Intensity Based on Multiple Linear Regression and Deep Learning

PONE-D-25-17894R2

Dear Dr. zhu,

We’re pleased to inform you that your manuscript has been judged scientifically suitable for publication and will be formally accepted for publication once it meets all outstanding technical requirements.

Kind regards,

Lingye Yao, Ph.D.

Academic Editor

PLOS One

Additional Editor Comments (optional):

Please carefully address the minor comment raised by the reviewer when you are doing the proofreading. "**Adding a list of urbanism methods inducing low carbon urbanism, in a couple of lines in the intro** ".

Reviewers' comments:

Reviewer's Responses to Questions

**Comments to the Author**

Reviewer #3: All comments have been addressed

2. Is the manuscript technically sound, and do the data support the conclusions?

Reviewer #3: Yes

3. Has the statistical analysis been performed appropriately and rigorously?

Reviewer #3: I Don't Know

4. Have the authors made all data underlying the findings in their manuscript fully available?

Reviewer #3: Yes

5. Is the manuscript presented in an intelligible fashion and written in standard English?

Reviewer #3: Yes

Reviewer #3: Yes, thanks for your reply. For me the article was already ok anyway. It was just a minor comment in case you were thinking to also add a list of urbanism methods inducing low carbon urbanism, in a couple of lines in the intro. But I was already satisfied also without such list.

**Do you want your identity to be public for this peer review?** For information about this choice, including consent withdrawal, please see our Privacy Policy

Reviewer #3: No

---

## [Editor Report · Acceptance letter]

PONE-D-25-17894R2

PLOS One

Dear Dr. zhu,

I'm pleased to inform you that your manuscript has been deemed suitable for publication in PLOS One. Congratulations! Your manuscript is now being handed over to our production team.

Kind regards,

on behalf of

Dr. Lingye Yao

Academic Editor

PLOS One